# Why health system diagnosis delay among tuberculosis patients in Illubabor, Oromia region, South West Ethiopia? A qualitative study

Jira Wakoya Feyisa[1], Jiregna Chalcisa Lemu[2], Megersa Dinku Hunde[2], Robera Demissie Berhanu🄳[2], Debela Dereje Jaleta🄳[2], Diriba Kumara Abdisa[3], Tadesse Sime Garedow[2], Keno Melkamu Kitila🄳[4]*

1 Department of Public Health, Institute of Health Sciences, Wollega University, Nekemte, Ethiopia,
2 Department of Nursing, College of Health Sciences, Mettu University, Mettu, Ethiopia, 3 Department of Health Informatics, College of Health Sciences, Mettu University, Mettu, Ethiopia, 4 Department of Public Health, College of Health Sciences, Mettu University, Mettu, Ethiopia

* qannoo63@gmail.com

**Data Availability Statement:** Data cannot be shared publicly because of the nature of the data being qualitative that contains personal quotes and

## Abstract

### Background

The main target of tuberculosis control and prevention is to detect incident cases as quickly as possible and also to prevent the occurrence of disease. It is also the responsibility of the health facility to screen the contacts, identifying children for prophylactic therapy. However, the target is difficult to meet due to issues within health facilities that cause health system diagnostic delays. Despite this, there is no information explored why health system diagnostic delays among tuberculosis patients.

### Method

A qualitative study was conducted by using different data collection methods and sources. Seventeen DOT providers, five laboratory professionals, six program managers, and seven Tuberculosis patients were interviewed. In addition, 22 governmental health facilities were observed for the availability of resources. Data obtained from the in-depth interviews was transcribed, coded, categorized, and thematized manually.

### Result

Health system Diagnosis delays reasons were related with sample collection procedures, poor competency of health professionals, in addition to absences or scarcity of health professionals, medical products, and basic infrastructure. We found 18 health facilities without skilled personnel in the OPD, 7 health facilities with a broken microscope, and almost all health facilities without a separate room for sputum examination. Furthermore, 12 (54.5%) and 14 (63.6%) health facilities lacked access to water and electric power, respectively.

clues to where the study occurred and can be potentially identifiable. However, data is available on request to the chair of Research and community Service Coordination Office (RCS) of Mettu University Faculty of Public Health and Medical Sciences (E-mail:mohammed.amin@meu.edu.et).

**Funding:** The authors received no specific funding for this work.

**Competing interests:** The authors have declared that no competing interests exist.

## Conclusion

Many reasons for TB diagnosis delays have been identified in the study area. Poor competence of health workers and scarcity of resources were identified. Depending on the finding, we suggest strengthening the health workers' training. Concrete strategies need to be designed to retain professionals. Training on human resource planning and budget preparation is needed for front-line managers. Managers have to work on the maintenance of diagnostic tools and facilitate transportation. The rural health facilities need to use another alternative power source.

## Introduction

Tuberculosis has been declared a global public health emergency for the past 25 years because it is the prominent reason of death from infectious disease among adults [1], with 7.1 million tuberculosis cases in 2019 [2]. Early diagnosis of TB cases are the cornerstones of global TB control initiatives [3]. Despite breakthroughs in knowledge regarding the importance of early case detection and treatment, the health system continues to miss millions of tuberculosis cases, resulting in long-term transmission and a rise in the pool of people at risk of contracting and developing the disease [4].

The gap between a patient's first visit to a health care facility with symptoms of tuberculosis and the diagnosis of tuberculosis is referred to as the health system diagnostic delay. Actually it will be considered as health system delay if it is not detected on the first visit [5]. Early diagnosis of tuberculosis cases has been identified as a key impediment to TB control, particularly in low-income countries such as Ethiopia [6]. It causes a severe acute phase of the disease. Furthermore, despite "free services" for TB, it results in TB patients suffering significant pre-diagnosis costs for care seeking [7–9].

According to the findings of a systematic review and meta-analysis, health system diagnosis delays ranged from 2 to 128.5 days, whereas they ranged from 6 to 28 days in other studies conducted in Ethiopia among tuberculosis patients [10, 11]. The number and type of providers encountered prior to TB diagnosis, educational status, level of severity, types of symptoms, and first visited health facility are all factors that contribute to health system diagnosis delay [9, 12–14].

The Ethiopian government has prioritized the control of tuberculosis (TB) and has placed TB prevention and control as one of the country's priority health initiatives in the country's Health Sector Transformation Plan (HSTP) [8]. In Ethiopia, TB case notification has been mostly based on passive and community-based enhanced TB case finding, with a focus on diagnosing TB among those who actively seek medical care with TB symptoms or who are recognized in the community by health extension workers. With this technique, the program was able to detect less than two-thirds of the annual projected TB cases, necessitating the implementation of additional but effective strategies in order to meet the lofty targets set by END TB by 2035 [15].

Furthermore, in order to improve timely case detection, care, and prevention of further transmission while maximizing resources, the national TB control program needed to re-orient the approach with an additional systematic and targeted screening strategy among close and household contacts of infectious TB cases, among high-risk populations, and for key populations where TB usually concentrates. Moreover, since 2013, Ethiopia has been establishing

a public-private-mix (PPM) initiative to include the private medical sector in providing TB services. However, only a tiny percentage of the private sector is participating [16].

Despite the growing number of quantitative studies on health system diagnosis delays, there is limited information on the causes of health system diagnosis delays explored qualitatively. As far as we know, no studies have been undertaken in the study area. Furthermore, contributing factors differ across societies, types of health facilities visited, and geographical regions between population groups in the same local settings and disease categories. Localized studies are needed to uncover area and population-specific characteristics linked to extended TB health system diagnosis delays. As a result, the goal of this study was to find out why health care providers in the Illubabor zone take so long to diagnose tuberculosis patients.

In the fight against tuberculosis, it's critical to have a good understanding of what causes health system diagnosis delays. The findings of this study could help in the development of treatments targeted at increasing early diagnosis of tuberculosis cases in order to commence effective chemotherapy and reduce the risk of disease transmission in the public as well as the burden of morbidity in patients. Similarly, the research can assist healthcare practitioners in providing appropriate information to improve TB diagnosis in time, particularly in the study areas.

Furthermore, the findings of this study can be used by the Illubabor Zone Health Department to build and develop a local plan and implementation strategy to reduce health system diagnosis delays. Finally, the study is important in that it provides information to government policymakers, program planners, and non-governmental organizations (NGOs) so that they can develop relevant interventional strategies. It may also encourage other researchers and policymakers to conduct more extensive research in this area.

## Methods and materials

### Study setting and period

The study took place in Illubabor Zone health facilities in the Oromia region, South-West Ethiopia, from March 15, 2021, to October 1, 2021. The zone is 600 kilometers from Ethiopia's capital city, Finfinnee/Addis Ababa. In 2021, there were 13 rural districts and one town administration in the Illubabora Zone. Currently, forty (40) public health facilities and five private clinics in the Illubabor zone provide DOT diagnosis and treatment.

### Sample size and sampling techniques

**In-depth interview.** Through a purposive selection using an intensity sampling technique, twenty-eight health professionals with at least six months of experience in diagnosing and treating tuberculosis, or leading a tuberculosis control program, were included in the study [5]. We picked this sampling technique because it allowed us to reach out to people with a lot of knowledge, allowing us to acquire a complete picture of the reasons behind tuberculosis diagnosis delays. Seventeen DOT providers, five laboratory experts, six program managers, and seven tuberculosis patients (a total of 35 key informants) were interviewed to learn more about the delays in tuberculosis diagnosis. These groups were chosen because they were thought to be knowledgeable about their jobs. When two or more DOT providers were present at the same location, the more experienced provider was chosen. Seven TB patients on treatment were selected for in-depth interviews from seven treatment sites using a "maximum variation" sampling technique, which allowed access to patients with a variety of background characteristics [17].

**Facility observation.** Seven districts and one town administration were chosen by the lottery method among 13 districts and one town administration [11]. Consequently, 22

governmental health facilities, including two hospitals, eight urban health centers (placed in districts or zonal towns), and 12 rural health centers were chosen from the sampled districts. Thus, different types of health facilities were included from a variety of districts to obtain a comprehensive picture of the presence or absence of necessary infrastructure and other resources for TB case diagnosis. To acquire a comprehensive picture of the presence or absence of necessary infrastructure and other resources for TB diagnosis, various types of health facilities from a variety of areas were included during observation.

## Data collection method and tool

Piloted interview guide questions were utilized for both the KIIs and the observations. We employed a variety of approaches to collect data. First, key informants in charge of directing or coordinating tuberculosis control programs at various levels, as well as those responsible for delivering DOT at the facility level and TB patients, were interviewed. All of the participants were approached face-to-face, and none of them declined to participate in the study. The inter-view guide was translated into Afan Oromo by an English language expert who is also an Afan Oromo native speaker and then re-translated into English.

The authors conducted in-depth interviews using the local language to learn about health workers' and patients' experiences and perspectives on the reasons for diagnosis delays. Various people were interviewed in order to gain as many opinions as possible on the study's ulti-mate goal. After transcribing the day's work and preliminary analysis, redundancy of responses was considered saturated and was deleted every evening.

The data gathering team consisted of Principal Investigators (male lecturers) with second-ary level qualifications (MPH). The study offered confidentiality to participants, and the inter-views took place in a private setting. We did our best to make the study participants feel at ease by engaging them in small conversation for the first few minutes. To ensure triangulation of the data with the record, the team captured the data and took notes, including memos of par-ticipant behaviour and context. The interviews were done as a dialogue rather than asking and answering questions. The interviews lasted 60 to 90 minutes and were audio-taped and tran-scribed verbatim, with descriptive field notes written after each of the interviews.

Second, governmental health facilities were inspected using a predetermined checklist to gain an overview of the current condition in terms of the availability of necessary infrastruc-ture and other resources, such as electricity, human resources, and TB diagnostic reagents. It was conducted by observing the TB room, laboratory room, outpatient department (OPD), patient waiting area, and general infrastructure, as well as interviewing the responsible individ-uals in the relevant facilities' units. Each facility's observation lasted 50 to 70 minutes. The authors were in charge of the facility inspection. The data gathering methods were created using national and WHO TB control program guidelines, as well as existing research [6–8].

## Data analysis, data quality control and trustworthiness

The data analysis for the in-depth interview began with daily transcription and preliminary analysis during the data collection phase. The writers transcribed all of the audio-taped inter-views and then double-checked the transcript by listening to the recordings. Descriptive notes were recorded when reviewing the transcripts. This included the addition of new probing questions as well as the consolidation of some of the interview guide's other questions. A health management specialist who is a native speaker of the local language (Afan Oromo) assessed de-identified audio-taped interviews for peer debriefing (transcripts).

The data coding and categorization were done manually by the authors. The material was identified and analyzed using a thematic analysis approach. The information was organized

into sample collection procedures, poor competency of health professionals, absences/scarcity of health professionals, scarcity of medical products/technologies, and absence/scarcity of basic infrastructures. Finally, we attempted to provide a general picture of what all of the survey participants described as their perspectives and experiences with regard to the reasons for TB diagnosis delays. The information was supplied in the form of descriptions of events and experiences that were pertinent to the study's goal [9].

The frequency of presence and absence of specified resources, as well as an acid fast bacilli (AFB) reagent stock-out, were checked and computed using facility observation data. Narrative data was also compiled based on observations of each health institution and its surroundings. As a result, the observations from the facilities were used to confirm the conclusions from the in-depth interviews, as well as to enhance and deepen the data interpretation.

Triangulating data sources and data collection methods helps to challenge data from a variety of sources. The information gathered from many people at various levels was used to obtain experiences of those working in the TB control program, as well as the patients who received care [10]. The interviewers (authors) also made every effort to ensure that their relationship with the study participants was founded on trust so that they felt safe.

The interviewers were familiar with the local languages and cultures, and they believed they had established an "optimal" (insider-outsider) distance from the study [11]. They considered themselves insiders because they spoke the native languages fluently, which helped them grasp the situation. By not collaborating with DOT providers and program administrators, the outsider status was kept in part.

## Ethics approval and consent to participate

Ethical approval was obtained from the Research and community Service Coordination Office (RCS) of Mettu University Faculty of Public Health and Medical Sciences with Ref. No RCS/ 114/13. Before the start of data collection, written informed consent was obtained from each of the study participants after information was provided about the nature and the study's goal. Participants were informed that they have the right not to participate in the study or can withdraw at any time. Furthermore, privacy and confidentiality were ensured by using numbers to identify and describe the study participants.

## Results

### Background characteristics of the study participants

From total 35 study participants took part in the in-depth interview, 28(80%) are health professionals, of which 23(82.8%) of them were male and 19(66.7%) of them were between the ages of 25 and 34. In terms of educational attainment, 20(72.8%) had a bachelor's degree, and 19 (68.2%) had > 5 years of work experience for their current position or responsibility (Table 1).

### Qualitative findings

From the analysis of the in-depth interviews, five major themes were identified. The major themes were: sample collection procedures, poor competency of health professionals, absences/ scarcity of health professionals, scarcity of medical products/technologies and absence/scarcity of basic infrastructures. Each was narrated with expressive quotes from participants.

### The sample collection method

Some laboratory professionals reported that the current approach to sputum collection might lead them to report false negative results. They explained that in the previous approach (spot-

**Table 1. Characteristics of the study participants for in-depth interviews in the Illubabor Zone, 2021.**

| Characteristics of respondents | Categories | Frequency | Percentage |
|---|---|---|---|
| Health workers(n = 28) | | | |
| Sex | Male | 23 | 82.8% |
| | Female | 5 | 17.2% |
| Age | ≤24 | - | |
| | 25–34 | 19 | 66.7% |
| | ≥35 | 9 | 33.7% |
| Education level | Diploma | 4 | 13.6% |
| | Bachelor degree | 20 | 72.8% |
| | Masters | 4 | 13.6% |
| Experience | <2 years | - | - |
| | Years | 9 | 31.8% |
| | >5 years | 19 | 68.2% |
| Patients(n = 7) | | | |
| Sex | Male | 3 | 42.9% |
| | Female | 4 | 57.1% |
| Age | <24 | 2 | 28.6% |
| | 24–34 | 2 | 28.6% |
| | >35 | 3 | 42.8% |
| Type of TB | Pulmonary TB | 6 | 85.7% |
| | Extra pulmonary TB | 1 | 14.3% |
| Duration of treatment | ≤2 months | 3 | 42.8% |
| | >2months | 4 | 57.2% |
| Residence | Urban | 5 | 71.4% |
| | **Rural** | **2** | **28.6%** |

morning-spot), morning sample had a high chance of getting the mycobacterium tuberculosis under the microscope. They mentioned that the morning sample was highly concentrated. In contrast to this, the current approach (spot-spot) is susceptible to collecting saliva only, which has a low probability of getting the mycobacterium tuberculosis under the microscope. One participant stated:" *We are collecting saliva, which is inappropriate sample.*"

Consequently, the participants suggested that more advanced research has to be done. According to their reports, the difference between the results of a sample from spot-spot and spot morning spot was significant. One participant said:"*. . . As to me, it is better to conduct sophisticated research.*"

## Poor competency of health professionals

Most of the study participants mentioned poor competency of health workers who were failing to request a necessary diagnostic investigation. They emphasized patients are suffering in the health facilities after consulting the health professionals.

One participant said:"*. . . in our health facility, diagnosis delays are happening because of mis-diagnosis at OPD level.*"

Additionally, respondents reported that the laboratory professionals' slide preparation led them to false negative reports. That means they are not preparing the slide with the appropriate procedure. One participant said:"*. . . for instance, if you observe their slide preparation, it is very poor in some health centers.*"

The participants (program managers, DOT providers, and laboratory professionals) described possible causes for the lack of health workers' competency. The first reason

mentioned was an absence of in-service TB training for all general practitioners, health officers, and laboratory professionals to provide quality TB services for their clients. One participant explained:". . . *concerning the quality starting from the OPD, they are not well updated with the new algorism method, including the general practitioner.”* Another participant said, "According to our health center, only one health professional was trained on tuberculosis from the outpatient department." *AS participants said if the managers think proactively, it is possible to give training for all health workers.*

The Second complaints by seniors about professionals’ competency were that higher education institutions are not producing competent graduates. They also underscored that the ministries of health and education have to take the necessary measures to correct the issues. One participant said:". . . compared *with the past, higher education institutions are producing professionals with unsatisfactory skills."*

The third reasons raised by the participants were non-utilization of national TB control program guideline. According to their idea among health professionals who have taken the training most of them are not doing accordingly. Even they have already dumped the guidelines to their home from which have taken the training. They are not doing according to guidelines which leading them to inappropriate history taking so that they miss right diagnosis and, miss diagnosis is the possible reason for diagnosing the other upper respiratory tract infections. Then prescribing antibiotics for the patients and patients back home without having the right diagnosis and treatment and then come back again after days or weeks for the patient is not relieved form his/her complaint. These not utilizing guidelines in turn explained due to the lack of close supervision.

Evidence from observation mentioned, that national TB control program(NTCP) guideline was not available in 16(72.7%) outpatient departments, 18(81.8%) laboratory rooms, and 12 (54.5%) TB rooms of the facilities.

## Absence/Scarcity of health professionals

A majority of the study participants among the DOT providers and program managers at various levels reported that there was a scarcity or absence of trained health professionals, particularly in the outpatient and laboratory departments, which caused TB diagnosis delay.

One participant explained, "The main reason for diagnosis delays happening in our health facilities is the scarcity of laboratory health professionals."

Moreover, our findings from observations at health facilities show that there was an absence of trained professionals at OPD in 18(81.8%) of the facilities. The respondents gave emphasis on the reason for health professional scarcity, which they mentioned as the cause of diagnosis delay. The scarcity of health professionals is related to high turnover as a result of joining nongovernmental institutions, education, and transfers to urban health facilities. The other reason for the delay in diagnosis is related to the absence of recruitment as per usual. The government is not recruiting health professionals immediately after graduation as a result of the scarcity of budgets. Additionally, significant numbers of university graduates are not passing the competency exam. According to the participant’s explanation, this condition is creating a huge gap in the midst of high professional demands.

One participant stated:". . . the *reason for health professionals’ scarcity could be the unemployment of university graduates immediately after graduation*."

Consequently, almost all the participants stated that high work overload is making it challenging to diagnose TB early. According to their reports, health professionals are not doing their work with adequate time because their intention was just to address the high patient flow.

One participant said, ". . . because of high patient flow, we are missing the opportunity to request an appropriate laboratory investigation, which is the reason for misdiagnosis."

One patient said, ". . . I visited three health facilities before my case was known; just as I entered the examination room, the health professionals gave me a prescription without asking me to address a lot of patients in the waiting area."

The participants reported that the work load was caused by the absence of the required health worker-to-population ratio standard by WHO. The DOT providers emphasized that their managers were ordering them to clerk more than sixty patients per day. This problem in turn happened as a result of not strictly requesting workers, and on the other hand, when the front line manager requests human power, the higher managers raise the budget scarcity issue. Accordingly, if the problem is not resolved, early tuberculosis diagnosis is out of the question, as per the participants.

## Scarcity of medical products/technologies

The absence of diagnostic tools like updated or functional microscopes, X-rays, functioning Gene expert machines, and a shortage of AFB reagents were major reasons for the delayed diagnosis of TB, as reported by many of the laboratory professionals and program managers.

One participant mentioned that "*the absence of diagnostic tools like X-ray, AFB reagents, and microscopes is the reason for the delay in diagnosis of TB.*"

From observation, in 7(31.8%) health facilities, microscopes were not functional and only 2 (9.1%) health facilities had X-rays. The AFB reagents were not available in the drugstores of most health facilities: acid alcohol in 15(68.2%), carbol-fuchsion in 17(77.3%), and methylene blue in 14(63.6%) health facilities on the day of observation. Functional weighing scales and masks were not present in 11(50%) and 13(59.1%) of the TB rooms of the facilities, respectively. Nevertheless, these resources are supposed to be available in all 22 health facilities.

According to participants, the possible causes of the shortage of reagents were a failure to timely distribute them to the health facilities, which was related to the absence of timely requests. Even though the reagents are prepared in decentralized local regions, the transportation mechanism does not maintain their quality. Additionally, participants were complaining about the lack of a vehicle or motorcycle for the laboratory department to bring the reagents to the Zonal level.

"*Most of the time, we are not supplied with AFB reagent on time, and even when it comes, they only supply us with local bottles that are not packed appropriately,*" one participant explained.

Participants also mentioned the cause of scarce diagnostic tools like microscopes and gene experts as the absence of timely maintenance for failed machines. One participant said, "*Now the time you are interviewing me, the Gen expert machine is not functioning, which is only found in hospitals and used for all health centers in the zone.*"

One patient undergoing directed observed treatment describes his ordeal: [. . .].*My result was delayed after I came to the health facility because of the machine not functioning, which is used for drug initiation.* [. . .](A TB patient)

Participants explained the lack of timely machine maintenance as not having trained technicians who maintain it. They stated that now they are calling for technicians from Addis to fix these machines. As a result, delays happen until the technician comes from a long distance. One participant stated: "We *are calling the technicians from Addis Abeba to fix the machines which are resulting in the tuberculosis diagnosis delay.*"

The other idea, according to the participants, was that most of the microscopes being used in their healthcare set up are out-dated. They emphasized that our country is not using

advanced technologies that are coming with new versions of machines. For this reason, they mentioned these out-dated machines may be the possible cause of the TB diagnosis delay. One participant mentioned that, "to *your surprise, most of the microscopes working in our health facilities are out-dated.*"

### Absence/Scarcity of basic infrastructure

Most of the participants mentioned the absence of adequate rooms, water supply, and electricity. Several rural health centers lacked clean water, electricity, a patient waiting area, and a sputum collection area.

From observation, there were no waiting areas for patients in 7(31.8%) of the health facilities. Of those who had patient waiting areas, none of the health facilities had a separate waiting area for TB patients. Moreover, no health centre had a sputum collection area, and only one hospital had a separate laboratory room for sputum examination. The 12(54.5%) and 8(36.4%) health facilities didn't have water supply and electricity, respectively.

The majority of study participants mentioned that limitations in laboratory room space and a separate sputum collection area were the causes of delayed diagnostics at health facilities. They said this could happen as a matter of poor building design. This led patients to give the samples using the same window. They said since limited space increases the chance of transmission for other patients, we appoint suspected cases after a certain time when patient flow decreases. One participant said, "*We are intentionally delaying the diagnosis of TB to decrease the chance of index case contact with other patients because they bring samples via one window.*"

Several participants gave emphasis to the absence of electricity and a standby generator as major factors in the diagnosis delay.

One participant explained, "*One day I had scheduled the suspected TB patients for the next day because there was no light.*"

One patient said: [. . .]. In fact, my results were delayed in the health facility due to appointments. [..]. (A TB patient)

According to the participants, the causes of limited basic infrastructure like water and electricity are scarcity of budget allocation and low community participation in fostering the basic infrastructure. Other issues identified were a lack of engagement with other aiding agencies or non-governmental organizations. Also, a lack of regular supervision was raised by participants. Most of them reported a lack of vehicles as the main obstacle to conducting supervision. Also, the COVID-19 impact was mentioned significantly. When the pandemic struck the country, most of the usual plans were disrupted.

### Discussion

The present study explored laboratory professionals', DOT providers', patients', and program managers' views and experiences related to health system diagnosis delays among TB patients in Southwest Ethiopia. Our findings suggest that there are different reasons for Health system diagnosis delays in the study area. The major reasons were issues related to sample collection methods, poor competency of health professionals, absences /scarcity of health professionals, absences or scarcity of medical products or technologies, and absences or scarcity of basic infrastructure.

As mentioned by laboratory professionals, the current sputum collection method (spot-spot) has been leading them to Health system diagnosis delay. The different studies' findings, including systematic review and meta-analysis, were against this finding [18–20]. Early morning sputum samples (EMS) are generally considered to yield a greater number of positive results than spot samples and to have higher sensitivity and specificity for culture, yet the

published data to support this assumption is scarce. Moreover, spot samples were found to be at least as good as EMS for identifying M. tuberculosis prior to and during TB treatment and do not support the superiority of EMS over spot samples in a clinical trial setting. Spot-spot sputum sample collection also reduces the number of visits by patients and helps in preventing drop-outs. Possible explanations related to the spot-spot sample collection method's delay might be the volume and quality of the sputum sample collected, the time to pro- cessing and transport conditions, and the proficiency of the laboratory professionals [19]. We propose that patients receive education before or during a spot sputum collection, including instructions on how to create sputum, advice on breathing exercises, and training for laboratory workers.

Poor competency of healthcare providers was another reason explained for contributing to health system diagnosis delays among tuberculosis patients. The discovery is supported by lab- oratory research [21, 22].The possible justifications are: skilled laboratory professionals can conduct diagnostic procedures correctly; competent health care providers working at OPD can take an appropriate history and do physical diagnosis accordingly so that he/she can request an accurate investigation. In this study, participants mentioned pre-service insufficient quality of education, a lack of all-inclusive in-service training, and not utilizing TB guidelines. Therefore, we suggest that health facilities at all levels should consider equipping health work- ers via training. Also, the ministries of health and education need to do more in collaboration to produce competent workers.

As mentioned by the TB control program coordinators, the health workers were diagnosing TB without special training on Tuberculosis. This finding is similar to studies conducted in different resource-limited countries [8, 23, 24]. They explained it as a result of turnovers of trained workers, which in turn happened because health workers transferred to the urban set- ting and were interested in joining non-governmental organizations. However, to accelerate progress towards universal health coverage and the UN Sustainable Development Goals, ensuring equitable access to health workers within strengthened health systems has planned strategic directions [25]. So, the scarcity of trained health care workers will have a negative effect on early TB diagnosis after patients visit health facilities [26]. This finding indicates that many things are expected from managers to make available health workers at all levels. For example, a sound retention strategy may motivate health workers to work in rural and remote health facilities.

Another reason mentioned for the scarcity of trained health professionals was the absence of timely recruitment, which is again due to budget scarcity and the need to request the needed health work force timely. This may also increase Health system diagnosis delays in health facil- ities. As a result, strategic health workforce management and the acquisition of new employees on a regular basis are needed [27]. For this reason, we suggest that training be given on human resource planning and budget preparation for the low-level managers so that such a problem can be tackled.

In Sub-Saharan countries like Ethiopia, most health facilities use AFB smear microscopy for TB diagnosis [22]. But, this study revealed that there were shortages of microscopies related to absence of maintenance, and similarly, the Gene Expert machines, which are being utilized for anti-TB initiation, were not working during the data collection period. Previous studies also reported similar findings [7, 8]. Because of malfunctioning microscopies, patients would be left without getting an accurate diagnosis; on the other hand, they might be referred to another health facility for laboratory investigation. Furthermore, if the Gene expert is no lon- ger able to provide an immediate solution, patients may be transferred to another hospital to begin anti-TB treatment. These conditions worsen TB diagnosis delays. In Ethiopia, the health sector transformation plan has also prioritized access to more sensitive screening tools, such as chest X-ray and GeneXpert, for diagnosing tuberculosis [28]. However, access to updated tools

(such as a fluorescence microscope) [29] and a lack of timely diagnostic machine maintenance are to blame. We suggest that the Federal Ministry of Health and Regional Health Bureau have to work on the maintenance of malfunctioning diagnostic tools. This could be realized by working with biomedical engineers so that they can avail themselves of professionals needed to maintain malfunctioning tools. In one or the other ways, the respective stakeholders sought to update the laboratory machines with time.

Laboratory professionals acknowledged interruptions of AFB reagents as a reason for diagnostic delays in health facilities. Similar findings are reported by different studies [14, 23, 29]. Due to the absence of AFB reagents, patients would be referred to hospitals and higher-level private health institutions where they could be available. The possible reasons mentioned by participants for not availing of reagents in a regular manner were a lack of a motorbike for this purpose and not requesting them on time. This implies the managers are expected to facilitate transport mechanisms and strengthen regular monitoring so that they can close the gap, increasing the chance of Health system diagnosis delay.

The current study has identified inadequate room to perform laboratory activities, an absence of electricity and running water. This finding is similar to studies done in other resource-limited countries [14, 24]. It is justifiable that using a microscope is not possible without electricity, which could have a high potential to create diagnostic delay. Even though Ethiopia has the goal of achieving universal access to electricity by 2025, only 44% of urban and 31% of rural Ethiopians have access to electricity [30]. In such a context, until the goal is achieved, we suggest that the rural health facility needs to use another alternative power source like solar energy, biogas, or generator.

Inadequate working space, the absence of a sputum collection area, and a separate laboratory room for sputum examination were identified as means of diagnosis delay in health facilities. This finding is also supported by previous studies [14]. The participants explained that delays can occur as a result of space constraints due to fears of cross-contamination among suspected patients and others. Laboratory professionals were postponing suspected patients for some hours after the other patients were seen. This scenario might lead the patients to dropout and return another time with a delayed case created as a result of a health facility problem. We propose that the existing health facilities need to be redesigned in order to curb the problem. So the strategic road map, which involves different stakeholders, needs to be developed.

As a limitation, the present study excluded the views of health extension workers. We interviewed patients who had visited public health facilities, but not those individuals who had visited private clinics, and observation was not done for private clinics. Thus, we might miss essential information regarding factors affecting diagnosis delays in private clinics. Moreover, the responses from the study participants might tend to be positive and may not fully address their concerns (social desirability bias) since most of the interviews were conducted at health facilities. We attempted to reduce this bias by informing them about the objective of the study, assuring their confidentiality and using indirect questioning, which meant asking about what others thought and felt. As strength of the study, we used in-depth interviews and facility observations. In addition, we have included health centers, primary hospitals, and specialized hospitals. This helped to provide a great number of data sources and triangulation methods.

## Conclusion

Many reasons for TB diagnosis delays have been identified in the study area. Poor competencies of health professionals, absences or scarcity of health professionals, scarcity of medical products or technologies, and absences or scarcity of basic infrastructures were identified. The

limited basic infrastructure identified included running water, electric power, and room. Scarce medical products were the absence of updated or functional microscope machines and timely maintenance of diagnostic tools.

Preventing delays in diagnosis of tuberculosis is a crucial measure in achieving the Stop Tuberculosis strategy. Therefore, we suggest strengthening the health workers' in-service and pre-service training. Concrete strategies need to be designed to acquire and maintain adequate and competent professionals. Training has to be given on human resource planning and budget preparation for the low-level managers. The Federal Ministry of Health and Regional Health Bureaus have to work on the maintenance of malfunctioning diagnostic tools, which could be realized in collaboration with biomedical engineers. The managers are expected to facilitate transport mechanisms to avail reagents and strengthen regular monitoring so that they can close the gap. Rural health care facilities must use an alternative power source such as solar energy, biogas, or generators. Finally, we propose that the existing health facilities need to be redesigned in order to curb the problem related to the room.

## Supporting information

**S1 Checklist. Checklist for facility observation.**
(PDF)

**S2 Checklist. Consolidated criteria for reporting qualitative research (COREQ) checklist.**
(DOCX)

**S1 Questionnaire. Interview guide English version.**
(PDF)

## Acknowledgments

We would like to thank the Illubabor Zone health department and respective district health offices. We would also like to thank the study participants and all others involved in any process of the study for providing us with the necessary support and information.

## Author Contributions

**Conceptualization:** Jira Wakoya Feyisa, Jiregna Chalcisa Lemu, Megersa Dinku Hunde, Keno Melkamu Kitila.

**Data curation:** Jira Wakoya Feyisa.

**Formal analysis:** Jira Wakoya Feyisa, Robera Demissie Berhanu, Debela Dereje Jaleta, Diriba Kumara Abdisa, Tadesse Sime Garedow, Keno Melkamu Kitila.

**Investigation:** Jira Wakoya Feyisa, Jiregna Chalcisa Lemu, Megersa Dinku Hunde, Keno Melkamu Kitila.

**Methodology:** Jira Wakoya Feyisa, Robera Demissie Berhanu, Debela Dereje Jaleta, Keno Melkamu Kitila.

**Writing – original draft:** Jira Wakoya Feyisa, Diriba Kumara Abdisa, Keno Melkamu Kitila.

**Writing – review & editing:** Jira Wakoya Feyisa, Keno Melkamu Kitila.

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
