## [Decision Letter · Decision Letter 0]

1 Jun 2022

PONE-D-22-11392Why Tuberculosis diagnosis and treatment delay in health system: a qualitative study of patients, health workers and program managers’ perspectivesPLOS ONE

Dear Dr. Kitila,

Thank you for submitting your manuscript to PLOS ONE. After careful consideration, we feel that it has merit but does not fully meet PLOS ONE’s publication criteria as it currently stands. Therefore, we invite you to submit a revised version of the manuscript that addresses the points raised during the review process.

We look forward to receiving your revised manuscript.

Kind regards,

Yasir Bin Nisar, Ph.D

Academic Editor

PLOS ONE

Journal Requirements:

Reviewers' comments:

Reviewer's Responses to Questions

**Comments to the Author**

1. Is the manuscript technically sound, and do the data support the conclusions?

Reviewer #1: Partly

Reviewer #2: Partly

2. Has the statistical analysis been performed appropriately and rigorously? 

Reviewer #1: N/A

Reviewer #2: N/A

3. Have the authors made all data underlying the findings in their manuscript fully available?

Reviewer #1: No

Reviewer #2: Yes

4. Is the manuscript presented in an intelligible fashion and written in standard English?

Reviewer #1: No

Reviewer #2: No

5. Review Comments to the Author

Reviewer #1: Dear Editor,

Thank you for giving me the opportunity to review the article “Why Tuberculosis diagnosis and treatment delay in health system: a qualitative study of patients, health workers and program managers’ perspectives”. I have read the article with great interest. However, major revisions are needed in the methods and writing style before it can be considered for publication. Some of the comments are as follows:

1. The article needs to be read and revised by a native English speaker.

2. The title of the article needs to be revised/rewritten.

3. Reviewer comment: In the introduction section and throughout the manuscript, definition of health system delay, “Early diagnosis and initiation of tuberculosis treatment” “Late detection and Treatment” are not clear.

4. Methods and Materials

Study setting and period

Study period has been written in the reverse order “from October 15, 2021, to March 1, 2021”.

5. Sample size and sampling techniques

In-depth interview

It is not mentioned that 28 health professionals were chosen out of how many health professionals?

6. Why was the criteria of 6 months of experience in TB for health professionals chosen?

7. What was the criteria for choosing different numbers of interviewees (Seventeen DOT providers, five laboratory experts, six program managers, and seven tuberculosis patients for the study?

8. Facility Observation

Who observed, what was their qualification, how frequently for how long they observed the facilities?

9. Interviewers

What was their qualification, were they trained in interviews and were their methods checked?

10. Finally, until the publication was approved,???? the authors had regular conversations and consultations about the design, data analysis, and result interpretations. Unclear and unnecessary information is given.

11. Ethics approval and consent to participate

Were the participants compensated for their time? Was any incentive given to the study participants?

12. The sample collection method is not clear regarding different collection procedures such as “spot spot” and “spot morning spot”techniques.

13. Criteria of trained professional in OPD has not been mentioned.

14. Discussion

First paragraph, No new finding has been reported as a result of the study. Scarcity of infrastructure is already known.

15. Were the skills of the lab professionals checked?

16. Conclusion is too long in abstract and end of discussion. Needs to be rewritten.

Best Regards

Brekhna

Reviewer #2: While this is an important research topic and very relevant for LMICs, this paper does not add any new information, nor is there any innovative idea or concept explored. It adds to the pool of the existing knowledge.

The paper has used purely qualitative methods, that include interviews and observations, as mentioned by authors. It would have been enriched had it been a mixed methods study, with some quantitative data on coverage, the duration between care seeking and confirmation of diagnosis, proportion of the total cases that were diagnosed at the first visit or required more than 1 visit, the data from facilities which were commonly visited by patients or any surveillance data from the community that collected information on TB prevalence.

Since it is purely a qualitative study, focus group discussions would have added greater insights into a larger number of key informants.

Additionally, as we know TB management has several challenges, the paper could have adopted a more comprehensive approach by addressing some more barriers if not all. Although the study has addressed the important barriers such as delays in diagnosis, lack of adherence to TB treatment guideline, poor infrastructure, lack of appropriate diagnostic tools, staff shortage etc., there are others, such as unregulated private health care leading to widespread irrational use of first-line and second-line anti-TB drugs; spreading HIV infection; poverty; lack of political will and administrative challenges. There are issues related to intermittent supplies of Anti-TB drugs, intermittent supplies of laboratory reagents, poor TB data documentation, lack of health worker motivation and commitment, lack of awareness, knowledge, stigma associated with the infection, long distance to health facilities limiting access, adverse drug reaction, and poor household income. Poor compliance to complete and appropriate treatment with a high proportion of defaulters, also leads to an increased pool of infective and drug resistant cases. There may be TB cases who do not report to hospital for care-seeking, due to lack of knowledge or fear of stigma, which also need urgent attention. There may be a need of active surveillance. It would be very useful to incorporate this additional information in case the study has collected these.

Coming to the specific comments, it would be helpful if the challenges could be categorized as 1) patient level 2) provider level 3) facility level 4) health systems level and 5) policy level. Currently there seems to be substantial overlap.

The term participants and lab professionals have been interchangeably used. It is not clear as which responses are provided by which category of respondents, whether it is reported by the program managers, or the DOT providers, or the lab experts or the patients. Table 1 needs to be categorized accordingly. We do not know the education of patients. It is also not clear why duration of treatment is categorized as less than and more than 2 months, what is the mean duration of treatment?

Finally, the English language needs improvement, there is lack of consistency in the grammar, use of language and spellings in the abstract, methods and materials and discussions; seems that the sections are written by different persons.

6. PLOS authors have the option to publish the peer review history of their article (what does this mean?). If published, this will include your full peer review and any attached files.

Reviewer #1: **Yes: **Brekhna Aurangzeb

Reviewer #2: **Yes: **Sarmila Mazumder

---

## [Author Response · Author response to Decision Letter 0]

23 Jun 2022

Response to Reviewers

Response to Reviewer 1

1. The article needs to be read and revised by a native English speaker.

Response: Yes, I have invited the English language expert to read and revise the manuscript. Consequently, the necessary improvement has been made.

2. The title of the article needs to be revised/rewritten

Response: Tuberculosis diagnosis delay: a qualitative study of patients, health workers, and program managers’ perspectives 

3. Reviewer comment: In the introduction section and throughout the manuscript, definition of health system delay, ‘’Early diagnosis and initiation of tuberculosis treatment’’ ‘’Late detection and Treatment’’ are not clear.

Response: Regarding TB delay, there are three terms.

1. Patient delays are defined as the time interval from the onset of symptoms of tuberculosis until the first visit to any formal health care facility above median days from the onset of the symptoms as cut-off points 

2. Health system delays: is the time interval from the patient’s first visit to the formal health care facility to the commencement of treatment, and time longer than the median days will be considered indicative of health system delays (diagnosis delay plus treatment delay).

3. Total health system delay is the time interval from the first onset of symptoms to the commencement of treatment, or the sum of patient delay and healthcare system delay.

In this manuscript, the title has focused on health system delays (diagnosis delay plus treatment delay). At the beginning, the authors' intention was to explore the reasons for health system delay (both reasons for diagnosis delay and treatment delay), but since all participants mentioned no reasons related to treatment delay, we have modified the title (see track change). Based on the comments We have modified the introduction section (see track change version).

4. Methods and Materials

Study setting and period

Study period has been written in the reverse order ‘’from October 15, 2021, to March 1, 2021’’.

Response: We have corrected the order accordingly (see line 68 from the track change).

5. Sample size and sampling technique

In-depth interview

It is not mentioned that 28 health professionals were chosen out of how many health professionals?

Response: Actually, Illubabor Zone has 13 districts and one town administration. Out of these, six districts and one town were selected randomly. As mentioned in the manuscript, the health professionals involved were program managers, laboratory professionals, and DOT providers. Regarding the six program managers, one was from the Zonal CDC focal person and the other five were from selected district health offices. Regarding the DOT providers, 17 were included out of 82 DOT providers in the seven districts. Five of 36 laboratory experts were from seven health facilities. 

6. Why was the criteria of 6 months of experience in TB for health professionals chosen?

Response: After six months, we expect any health professional to know their work environment. In Ethiopia, they can give enough information about the reasons for the TB diagnosis delay. 

7. What was the criteria for choosing different numbers of interviewees(Seventeen DOT providers, five laboratory experts, six program)

Response: For triangulation, different numbers of health professionals were interviewed. 

8. Facility Observation

Who observed, what was their qualification, how frequently for how long they observed the facilities?

Response: At each facility, observations were conducted for 50 to 70 minutes by the authors. They had masters in educational qualifications (see 124 & 125 lines from the track change version). 

9. Interviewers

What was their qualification, were they trained in interviews and were their methods checked?

Response: The interviewers were the authors (originators of the research idea).We had a common understanding of data collection procedures. 

10. Finally until the publication was approved? ??? the authors had regular conversations and consultations about the design, data analysis, and result interpretations. Unclear and unnecessary information is given.

 Response: We have removed it. See track change version. 

11. Ethics approval and consent to participate 

Were the participants compensated for their time?

Was any incentive given to the study participants?

Response: Any incentives were not given to participants. 

12. The sample collection method is not clear regarding different collection procedures such as ‘’spot spot’’ and ‘’spot morning spot’’ techniques.

Response: A spot morning spot technique is for sputum microscopy. One spot sample is collected at the time of the first visit of a patient to the laboratory. Two sputum (spot and early morning) samples were collected the next day.

In 2011, WHO advice was revised with a recommendation of a two-‘spot-spot’ strategy collected on the same day. According to the participants, after the TB diagnosis technique changed from spot morning spot to spot-spot it has been contributing to the diagnosis delay. 

13. A criteria of trained professional in OPD has not been mentioned.

Response: Any health professional who has taken any type of in-service TB training is considered a trained health professional. 

14. Discussion

First paragraph, No new finding has been reported as a result of the study. Scarcity of infrastructure is already known.

Response: OK, This research topic is new. Because previous studies were conducted quantitatively and others were conducted on factors affecting case identification. But, this research explored reasons for diagnosis delays qualitatively. In particular, it is helpful for the study area. Because contributing factors differ across societies, types of health facilities visited, and geographical regions between population groups in the same local settings and disease categories. Localized studies are needed to uncover area and population-specific characteristics linked to extended TB diagnosis delays. 

15. Were the skills of the lab professionals checked?

Response: Dear Reviewer, The study's intention is not to measure (check) the skills of lab professionals. We have just recorded factors delaying TB diagnosis by participants.

16. Conclusion is too long in abstract and end of discussion. Needs to be rewritten.

Response: Thank you. It has been modified accordingly; see track change version. 

Response to Reviewer 2

1st paragraph comment: While this is an important research topic and very relevant for LMICs, this paper does not add any new information, nor is there any innovative idea or concept explored. It adds to the pool of the existing knowledge.

Response: Thank you for your critical comments. But, the previous studies focused on factors affecting case detection. In this study, we focused on factors affecting TB diagnosis delay. That means factors which could delay diagnosis after the patients reach health facilities.

2nd paragraph comment: The paper has used purely qualitative methods,that include interviews and observations, as mentioned by authors. It would have been enriched had it been a mixed methods study, with some quantitative data on coverage, the duration between care seeking and confirmation of diagnosis, proportion of the total cases that were diagnosed at the first visit or required more than 1 visit, the data from facilities which were commonly visited by patients or any surveillance data from the community that collected information on TB prevalence.

Response: The quantitative studies were conducted by other researchers who measured the duration between care seeking and confirmation of diagnosis. But, in this study, our aim was to explore qualitatively "why TB diagnosis delays after patients reach the health facilities." 

3rd comment: Since it is purely a qualitative study, focus group discussions would have added greater insights into a larger number of key informants.

Response: As we have mentioned in the method section, the participants were included from different districts, which are from distant geographic locations from each other’s. So, it was difficult to bring them together for focus group discussion, and we preferred interviewing them individually. 

4th paragraph comment: Additionally, as we know TB management has several challenges, the paper could have adopted a more comprehensive approach by addressing some more barriers if not all. Although the study has addressed the important barriers such as delays in diagnosis, lack of adherence to TB treatment guideline, poor infrastructure, lack of appropriate diagnostic tools, staff shortage etc., there are others, such as unregulated private health care leading to widespread irrational use of first-line and second-line anti-TB drugs; spreading HIV infection; poverty; lack of political will and administrative challenges. There are issues related to intermittent supplies of Anti-TB drugs, intermittent supplies of laboratory reagents, poor TB data documentation, lack of health worker motivation and commitment, lack of awareness, knowledge, stigma associated with the infection, long distance to health facilities limiting access, adverse drug reaction, and poor household income. Poor compliance to complete and appropriate treatment with a high proportion of defaulters, also leads to an increased pool of infective and drug resistant cases. There may be TB cases who do not report to hospital for care-seeking, due to lack of knowledge or fear of stigma, which also need urgent attention. There may be a need of active surveillance. It would be very useful to incorporate this additional information in case the study has collected these.

Response: Dear Reviewer, the above reasons could be an answer to "why low TB case notification". As a result, it covers a wide range of topics, including barriers to TB diagnosis before and after patients visit health facilities. In our case, we focused on delay of TB diagnosis. That means when our concern was on delay time of diagnosis whereas the above might be focused on low case detection.

5th comment: Coming to the specific comments, it would be helpful if the challenges could be categorized as 1) patient level 2) provider level 3) facility level 4) health systems level and 5) policy level. Currently there seems to be substantial overlap.

Response: The proposed categories are good. But, based on the collected data, the procedure of thematic qualitative analysis leads us to the thematic mentioned in the result section. Moreover, when we evaluated our data to adjust according to the comments, some categories couldn’t be identified from the data, like patient level, health system level, and policy level.

Comment: The term participants and lab professionals have been interchangeably used. It is not clear as which responses are provided by which category of respondents, whether it is reported by the program managers, or the DOT providers, or the lab experts or the patients. Table 1 needs to be categorized accordingly. 

Response: In the manuscript, where we have used "health professionals," it represents the ideas of all health professionals, but where we have used "lab professionals," it represents only the ideas of lab professionals. 

Comment: We do not know the education of patients. It is also not clear why duration of treatment is categorized as less than and more than 2 months, what is the mean duration of treatment?

Response: TB treatment phases are categorized into two phases. Duration of treatment less than 2 months is categorized as an intensive phase, whereas duration of treatment greater than 2 months is categorized as a continuation phase. 

Comment: Finally, the English language needs improvement, there is lack of consistency in the grammar, use of language and spellings in the abstract, methods and materials and discussions; seems that the sections are written by different persons.

Response: Yes, we have improved it accordingly.

---

## [Decision Letter · Decision Letter 1]

2 Sep 2022

PONE-D-22-11392R1Tuberculosis diagnosis delay: a qualitative study of patients, health workers and program managers’ perspectivesPLOS ONE

Dear Dr. Kitila,  Thank you for submitting your manuscript to PLOS ONE. After careful consideration, we feel that it has merit but does not fully meet PLOS ONE’s publication criteria as it currently stands. Therefore, we invite you to submit a revised version of the manuscript that addresses the points raised during the review process.

We look forward to receiving your revised manuscript.

Kind regards,

Yasir Bin Nisar, Ph.D

Academic Editor

PLOS ONE

Reviewers' comments:

Reviewer's Responses to Questions

**Comments to the Author**

1. If the authors have adequately addressed your comments raised in a previous round of review and you feel that this manuscript is now acceptable for publication, you may indicate that here to bypass the “Comments to the Author” section, enter your conflict of interest statement in the “Confidential to Editor” section, and submit your "Accept" recommendation.

Reviewer #1: All comments have been addressed

2. Is the manuscript technically sound, and do the data support the conclusions?

Reviewer #1: Yes

3. Has the statistical analysis been performed appropriately and rigorously? 

Reviewer #1: Yes

4. Have the authors made all data underlying the findings in their manuscript fully available?

Reviewer #1: No

5. Is the manuscript presented in an intelligible fashion and written in standard English?

Reviewer #1: No

6. Review Comments to the Author

Reviewer #1: Overall comments;

It is an important article that explains qualitative aspects of delay in diagnosis of TB in low middle income countries. The authors have revised it significantly. However, I still feel the English language and expression needs improvement at multiple points. There are multiple typo errors: in some places there is a space between bracket and figures and in certain places without a space, which needs to be corrected. Please add the revision of methods suggested by the reviewers in the manuscript.

Abstract:

Comment No 1:

Background: “The main target of Tuberculosis control and prevention is to detect incident cases as early as possible”. However, the target is hard to meet due to problems that exist within health facilities, which lead to diagnostic delays. Despite this, there is no information explored why diagnosis delays among Tuberculosis patients.” This statement is not correct. The main target of TB control and prevention is not only to detect incident cases as quickly as possible but also to prevent occurrence of disease. This is also the responsibility of the health facility to screen the contacts, identify cases and prevent spread in children by identifying children for prophylactic therapy.

Some examples of the problems stated above are;

Comment No 2

Results: Diagnosis delays have been identified as a result of issues with sample collection procedures, poor competency of health professionals, absences/scarcity of health professionals, and scarcity of medical products and absence/scarcity of basic infrastructure. This sentence can be rephrased to avoid repetition of absence/ scarcity.

Comment No 3; Result: “We found 18 health facilities without skilled personnel in the OPD, seven health facilities with a broken microscope, and almost all health facilities without a

separate room for sputum examination”. Grammatical correction needs to be done 7 health facilities had a broken microscope?

Comment no 4: Conclusion: Many reasons for TB diagnosis delays have been identified in the study area. Poor competence of health workers and scarcity of resources were identified. Depending on the finding, we suggest strengthening the health workers’ training. Concrete strategies need to be designed to retain Professionals. Training on human resource planning and budget preparation is needed for low level managers. The meaning of low level managers is not clear.

Introduction:

Comment No 5: Introduction line 3; why expected 10.4 million cases? When the year is already complete and should have the actual figures. Also, why has the statistics of 2016 been mentioned instead of the latest? Please put the latest statistics.

Comment No 6; “The time elapsed between seeking health care and receiving a diagnosis is defined as diagnosis delay” Why is definition in the past tense in introduction?

Comment No 7; “Diagnosis of tuberculosis cases has been identified as a key impediment

to TB control, particularly in low-income countries such as Ethiopia”. Probably the author wants to state that early diagnosis or prompt diagnosis is a key impediment to ......

Comment No 8; “Because, it causes more acute sickness, a longer period of infection, and worse treatment outcomes such as mortality and drug resistance”. Please replace with it causes severe acute phase of the disease.

Comment No 9:

Furthermore, despite "free service" for TB, it results in TB patients suffering significant pre-diagnosis costs for care seeking. Add s to service.

Comment No 10:

According to the findings of a systematic review and meta-analysis, diagnosis delays ranged

from 2 to 128.5 days, whereas they ranged from 6 to 28 days in other studies conducted in

Ethiopia among tuberculosis patients (10, 11). Is 2 or 6 days counted as diagnostic delay or is it normal time taken to diagnose TB????

7. PLOS authors have the option to publish the peer review history of their article (what does this mean?). If published, this will include your full peer review and any attached files.

Reviewer #1: **Yes: **Brekhna Aurangzeb

While revising your submission, please upload your figure files to the Preflight Analysis and Conversion Engine (PACE) digital diagnostic tool, https://pacev2.apexcovantage.com/. PACE helps ensure that figures meet PLOS requirements. To use PACE, you must first register as a user. Registration is free. Then, login and navigate to the UPLOAD tab, where you will find detailed instructions on how to use the tool. If you encounter any issues or have any questions when using PACE, please email PLOS at figures@plos.org. Please note that Supporting Information files do not need this step

---

## [Author Response · Author response to Decision Letter 1]

8 Sep 2022

Response to Reviewers

Reviewer #1: Overall comments;

It is an important article that explains qualitative aspects of delay in diagnosis of TB in low middle income countries. The authors have revised it significantly. However, I still feel the English language and expression needs improvement at multiple points. There are multiple typo errors: in some places there is a space between bracket and figures and in certain places without a space, which needs to be corrected. Please add the revision of methods suggested by the reviewers in the manuscript.

Response: Thank you. We have modified accordingly.

Abstract:

Comment No 1:

Background: “The main target of Tuberculosis control and prevention is to detect incident cases as early as possible”. However, the target is hard to meet due to problems that exist within health facilities, which lead to diagnostic delays. Despite this, there is no information explored why diagnosis delays among Tuberculosis patients.” This statement is not correct. The main target of TB control and prevention is not only to detect incident cases as quickly as possible but also to prevent occurrence of disease. This is also the responsibility of the health facility to screen the contacts, identify cases and prevent spread in children by identifying children for prophylactic therapy.

Response: Yes, it has been corrected accordingly. see first paragraph on page 2.

Some examples of the problems stated above are;

Comment No 2

Results: Diagnosis delays have been identified as a result of issues with sample collection procedures, poor competency of health professionals, absences/scarcity of health professionals, and scarcity of medical products and absence/scarcity of basic infrastructure. This sentence can be rephrased to avoid repetition of absence/ scarcity.

Response: Yes, it has been rephrased accordingly. See the result part of abstract.

Comment No 3; Result: “We found 18 health facilities without skilled personnel in the OPD, seven health facilities with a broken microscope, and almost all health facilities without a

separate room for sputum examination”. Grammatical correction needs to be done 7 health facilities had a broken microscope?

Response: Yes, it has been rephrased accordingly. See the result part of abstract. 

Comment no 4: Conclusion: Many reasons for TB diagnosis delays have been identified in the study area. Poor competence of health workers and scarcity of resources were identified. Depending on the finding, we suggest strengthening the health workers’ training. Concrete strategies need to be designed to retain Professionals. Training on human resource planning and budget preparation is needed for low level managers. The meaning of low level managers is not clear.

Response: Low level managers mean frontline managers. Now we have replaced with frontline managers. Check the last line of conclusion in the Abstract

Introduction:

Comment No 5: Introduction line 3; why expected 10.4 million cases? When the year is already complete and should have the actual figures. Also, why has the statistics of 2016 been mentioned instead of the latest? Please put the latest statistics.

Response: we have updated the information. check second page, line three

Comment No 6; “The time elapsed between seeking health care and receiving a diagnosis is defined as diagnosis delay” Why is definition in the past tense in introduction?

Response: We have modified accordingly. Check second paragraph, line one under the introduction.

Comment No 7; “Diagnosis of tuberculosis cases has been identified as a key impediment

to TB control, particularly in low-income countries such as Ethiopia”. Probably the author wants to state that early diagnosis or prompt diagnosis is a key impediment to ......

Response: You are right. We have corrected accordingly.

Comment No 8; “Because, it causes more acute sickness, a longer period of infection, and worse treatment outcomes such as mortality and drug resistance”. Please replace with it causes severe acute phase of the disease.

Response: Yes, we have replaced accordingly. check second paragraph, fourth line under the introduction.

Comment No 9:

Furthermore, despite "free service" for TB, it results in TB patients suffering significant pre-diagnosis costs for care seeking. Add s to service.

Response: Good. it is corrected.

Comment No 10:

According to the findings of a systematic review and meta-analysis, diagnosis delays ranged

from 2 to 128.5 days, whereas they ranged from 6 to 28 days in other studies conducted in

Ethiopia among tuberculosis patients (10, 11). Is 2 or 6 days counted as diagnostic delay or is it normal time taken to diagnose TB????

Response: Actually the ranges are mentioned to show the existing days of delay from the literature. But different literatures utilized the median as cut point.

---

## [Editor Report · Decision Letter 2]

26 Sep 2022

PONE-D-22-11392R2Tuberculosis diagnosis delay: a qualitative study of patients, health workers and program managers’ perspectivesPLOS ONE

Dear Dr. Kitila, 

Thank you for submitting your manuscript to PLOS ONE. After careful consideration, we feel that it has merit but does not fully meet PLOS ONE’s publication criteria as it currently stands. Therefore, we invite you to submit a revised version of the manuscript that addresses the points raised during the review process.

We look forward to receiving your revised manuscript.

Kind regards,

Yasir Bin Nisar, Ph.D

Academic Editor

PLOS ONE

Additional Editors Comments:

Thank you for giving me the opportunity to review the article “Why Tuberculosis diagnosis and treatment delay in health system: a qualitative study of patients, health workers and program managers’ perspectives”. I have read the article with great interest. However, major revisions are needed in the methods and writing style before it can be considered for publication. Some of the comments are as follows:

1. The article needs to be read and revised by a native English speaker.

2. The title of the article needs to be revised/rewritten.

3. Reviewer comment: In the introduction section and throughout the manuscript, definition of health system delay, “Early diagnosis and initiation of tuberculosis treatment” “Late detection and Treatment” are not clear.

4. Methods and Materials

Study setting and period

Study period has been written in the reverse order “from October 15, 2021, to March 1, 2021”.

5. Sample size and sampling techniques

In-depth interview

It is not mentioned that 28 health professionals were chosen out of how many health professionals?

6. Why was the criteria of 6 months of experience in TB for health professionals chosen?

7. What was the criteria for choosing different numbers of interviewees (Seventeen DOT providers, five laboratory experts, six program managers, and seven tuberculosis patients for the study?

8. Facility Observation

Who observed, what was their qualification, how frequently for how long they observed the facilities?

9. Interviewers

What was their qualification, were they trained in interviews and were their methods checked?

10. Finally, until the publication was approved,???? the authors had regular conversations and consultations about the design, data analysis, and result interpretations. Unclear and unnecessary information is given.

11. Ethics approval and consent to participate

Were the participants compensated for their time? Was any incentive given to the study participants?

12. The sample collection method is not clear regarding different collection procedures such as “spot spot” and “spot morning spot”techniques.

13. Criteria of trained professional in OPD has not been mentioned.

14. Discussion

First paragraph, No new finding has been reported as a result of the study. Scarcity of infrastructure is already known.

15. Were the skills of the lab professionals checked?

16. Conclusion is too long in abstract and end of discussion. Needs to be rewritten.

---

## [Author Response · Author response to Decision Letter 2]

3 Oct 2022

Response to Reviewers

1. The article needs to be read and revised by a native English speaker.

Response: Yes, I have invited the English language expert to read and revise the manuscript. Consequently, the necessary improvement has been made.

2. The title of the article needs to be revised/rewritten

Response: Of course, revised. See the track change 

3. Reviewer comment: In the introduction section and throughout the manuscript, definition of health system delay, ‘’Early diagnosis and initiation of tuberculosis treatment’’ ‘’Late detection and Treatment’’ are not clear.

 Response: see the second paragraph of track change under the introduction part.

4. Methods and Materials

Study setting and period

Study period has been written in the reverse order ‘’from October 15, 2021, to March 1, 2021’’.

Response: We have corrected the order accordingly (see study setting and period subsection).

5. Sample size and sampling technique

In-depth interview

It is not mentioned that 28 health professionals were chosen out of how many health professionals?

Response: Actually, Illubabor Zone has 13 districts and one town administration. Out of these, six districts and one town were selected randomly. As mentioned in the manuscript, the health professionals involved were program managers, laboratory professionals, and DOT providers. Regarding the six program managers, one was from the Zonal CDC focal person and the other five were from selected district health offices. Regarding the DOT providers, 17 were included out of 82 DOT providers in the seven districts. Five of 36 laboratory experts were from seven health facilities. 

6. Why was the criteria of 6 months of experience in TB for health professionals chosen?

Response: After six months, we expect any health professional, know their work environment. They can give enough information about the reasons for the TB diagnosis delay. 

7. What was the criteria for choosing different numbers of interviewees(Seventeen DOT providers, five laboratory experts, six program)

Response: For triangulation, different numbers of health professionals were interviewed. 

8. Facility Observation

Who observed, what was their qualification, how frequently for how long they observed the facilities?

Response: At each facility, observations were conducted for 50 to 70 minutes by the authors. They had masters in educational qualifications (see 124 & 125 lines from the track change version). 

9. Interviewers

What was their qualification, were they trained in interviews and were their methods checked?

Response: The interviewers were the authors (originators of the research idea).We had a common understanding of data collection procedures. 

10. Finally until the publication was approved? ??? the authors had regular conversations and consultations about the design, data analysis, and result interpretations. Unclear and unnecessary information is given.

 Response: We have removed it. See track change version. 

11. Ethics approval and consent to participate 

Were the participants compensated for their time?

Was any incentive given to the study participants?

Response: Any incentives were not given to participants. 

12. The sample collection method is not clear regarding different collection procedures such as ‘’spot spot’’ and ‘’spot morning spot’’ techniques.

Response: A spot morning spot technique is for sputum microscopy. One spot sample is collected at the time of the first visit of a patient to the laboratory. Two sputum (spot and early morning) samples were collected the next day.

In 2011, WHO advice was revised with a recommendation of a two-‘spot-spot’ strategy collected on the same day. According to the participants, after the TB diagnosis technique changed from spot morning spot to spot-spot it has been contributing to the diagnosis delay. 

13. A criteria of trained professional in OPD has not been mentioned.

Response: Any health professional who has taken any type of in-service TB training is considered a trained health professional. 

14. Discussion

First paragraph, No new finding has been reported as a result of the study. Scarcity of infrastructure is already known.

Response: OK, This research topic is new. Because previous studies were conducted quantitatively and others were conducted on factors affecting case identification. But, this research explored reasons for diagnosis delays qualitatively. In particular, it is helpful for the study area. Because contributing factors differ across societies, types of health facilities visited, and geographical regions between population groups in the same local settings and disease categories. Localized studies are needed to uncover area and population-specific characteristics linked to extended TB diagnosis delays. 

15.Were the skills of the lab professionals checked?

Response: Dear Reviewer, The study's intention is not to measure (check) the skills of lab professionals. We have just recorded factors delaying TB diagnosis by participants.

16.Conclusion is too long in abstract and end of discussion. Needs to be rewritten.

Response: Thank you. It has been modified accordingly; see unmarked version.

---

## [Decision Letter · Decision Letter 3]

21 Nov 2022

Why Health System Diagnosis Delay among Tuberculosis patients in Illubabor, Oromia Region, South West Ethiopia? A qualitative study

PONE-D-22-11392R3

Dear Dr. Kitila,

We’re pleased to inform you that your manuscript has been judged scientifically suitable for publication and will be formally accepted for publication once it meets all outstanding technical requirements.

Kind regards,

Yasir Bin Nisar, Ph.D

Academic Editor

PLOS ONE

Additional Editor Comments (optional):

Reviewers' comments:

Reviewer's Responses to Questions

**Comments to the Author**

1. If the authors have adequately addressed your comments raised in a previous round of review and you feel that this manuscript is now acceptable for publication, you may indicate that here to bypass the “Comments to the Author” section, enter your conflict of interest statement in the “Confidential to Editor” section, and submit your "Accept" recommendation.

Reviewer #1: All comments have been addressed

2. Is the manuscript technically sound, and do the data support the conclusions?

Reviewer #1: Yes

3. Has the statistical analysis been performed appropriately and rigorously? 

Reviewer #1: Yes

4. Have the authors made all data underlying the findings in their manuscript fully available?

Reviewer #1: Yes

5. Is the manuscript presented in an intelligible fashion and written in standard English?

Reviewer #1: Yes

6. Review Comments to the Author

Reviewer #1: All the comments have been addressed satisfactorily. This article should be published as it addresses a pertinent issue .

7. PLOS authors have the option to publish the peer review history of their article (what does this mean?). If published, this will include your full peer review and any attached files.

Reviewer #1: **Yes: **Brekhna Aurangzeb

---

## [Editor Report · Acceptance letter]

23 Nov 2022

PONE-D-22-11392R3 

Why Health System Diagnosis Delay among Tuberculosis patients in Illubabor, Oromia Region, South West Ethiopia? A qualitative study 

Dear Dr. Kitila:

I'm pleased to inform you that your manuscript has been deemed suitable for publication in PLOS ONE. Congratulations! Your manuscript is now with our production department. 

Kind regards, 

on behalf of

Dr. Yasir Bin Nisar 

Academic Editor

PLOS ONE